# Exploring alternative cytokines as potential biomarkers for latent tuberculosis infection in pregnant women

**Agnes Rengga Indrati**[1]*, **Anton Sumarpo**[1,2], **Petty Atmadja**[1], **Rositha Ratna Wisesa**[1], **Mohammad Ghozali**[3], **Raden Tina Dewi Judistiani**[4], **Budi Setiabudiawan**[5]

**1** Department of Clinical Pathology, Faculty of Medicine, Universitas Padjadjaran/Hasan Sadikin Central Hospital, Bandung, Indonesia, **2** Department of Clinical Pathology, Faculty of Medicine, Maranatha Christian University, Bandung, Indonesia, **3** Department of Biomedical Sciences, Faculty of Medicine, Universitas Padjadjaran/Hasan Sadikin Central Hospital, Bandung, Indonesia, **4** Department of Public Health, Faculty of Medicine, Universitas Padjadjaran, Bandung, Indonesia, **5** Department of Pediatrics, Faculty of Medicine, Universitas Padjadjaran/Hasan Sadikin Central Hospital, Bandung, Indonesia

* agnes.indrati@unpad.ac.id

**Data Availability Statement:** All relevant data are within the paper and its Supporting information files.

## Abstract

### Background

Interferon gamma release assays (IGRAs) are widely used to determine latent tuberculosis infection status. However, its pregnancy-affected performance and cost-expensive nature warrants for different alternatives for pregnant women. This study aims to evaluate the diagnostic performance of several alternative cytokines, including interleukin 2 (IL-2), interleukin 10 (IL-10), and interferon gamma-induced protein 10 (IP-10) to identify latent tuberculosis status in pregnant women.

### Materials and methods

123 pregnant womens were recruited for this study. The IGRA status was determined by using QuantiFERON Gold In-Tube. Meanwhile, we measured the level IL-2, IL-10, and IP-10 by using sandwich-microELISA method. We performed normality and comparison test by SPSS. In addition, receiver-operator characteristic (ROC) analyses and the optimal cut-off scores were identified using the EasyROC webtool.

### Results

We showed that IL-2, IL-10, and IP-10 were able to discriminate between IGRA-negative and IGRA-positive pregnant women. Moreover, IP-10 showed the highest discriminatory and diagnostic performance when compared to IL-2 and IL-10 with area under the curve (AUC) of 0.96 and cutoff point of 649.65 pg/mL.

### Conclusions

Our study showed that IP-10 can be considered as a promising alternative biomarker for IGRAs to diagnose LTBI in pregnant women.

**Funding:** - Initials of the authors who received each award: ARI - Grant numbers awarded to each author: 1959/UN6.3.1/PT.00/2021 - The full name of each funder: Universitas Padjadjaran - URL of each funder website: http://media.unpad.ac.id/files/kontrak/2021/hru/kontrak_8252.pdf The funders had no role in study design, data collection and analysis, decision to publish, or preparation of the manuscript.

**Competing interests:** The authors have declared that no competing interests exist.

## Introduction

Tuberculosis remains a major health burden in the world with an estimation of 10 million people developed active disease [1]. Among them, approximately 200.000 pregnant women were diagnosed with an active TB disease, where Southeast-Asian region is considered as one of the greatest burden [2]. Strikingly, 900 million pregnant women globally are estimated to have latent TB infection (LTBI), a condition where human immune responses to *Mycobacterium tuberculosis* (MTB) infection are kept active even without clinical manifestation, radiological abnormalities, or microbiological evidences of active TB infection [2–6].

There are two most commonly used methods to detect LTBI are the tuberculin skin test (TST) and interferon gamma release assays (IGRAs) [7]. TST works on the principle of delayed-type hypersensitivity skin reactivation to tuberculin purified protein derivate, meanwhile IGRAs was developed to detect the interferon gamma (IFN-γ) production by sensitized T cells on the presence of *M. tuberculosis* antigen [8, 9]. Of note, IGRAs might be more preferable due to the utilization of 6-kDa early secretory antigenic target (ESAT-6) and the 10-kDa culture filtrate protein (CFP-10) encoded in the region of differentiation 1 (RD1) present in *M. tuberculosis* but absent in Bacillus Calmette–Guerin (BCG) vaccine and non-tuberculous mycobacteria (NTM) [9, 10]; however, IGRAs are dependable on technical performance and relatively more expensive than TST, especially for middle- and low-income countries [11]. In addition, pregnancy-related immune modification was reported to influence the performance of both TST and IGRAs [12–14]. Consequently, IGRA testing might be unreliable in physiological immunosuppresive settings as in pregnancy. Moreover, the immune modification event in pregnancy could potentially initiate bacterial replication and eventually promote active TB development in pregnant women with LTBI [15]. Therefore, investigating potentially easy-to-implement, low-cost and highly reliable cytokine biomarkers for LTBI in pregnant women is a mandatory process.

One potential candidate is the interferon-γ inducible protein 10 (IP-10) [15–17]. IP-10, a member of CXC chemokine family, is expressed by antigen-presenting cells in concordance to the release of interferon- γ (IFN-γ) and regulate T-cell migration through the binding to CXC chemokine receptor 3 (CXCR3) [18, 19]. IP-10 has been assessed as a potential biomarker for LTBI due to less dependancy of its expression on cell-mediated immune capacity, thus its level might be consistent in pregnant women with LTBI [17, 20]. Similarly, interleukin-2 (IL-2) is a specific cytokine produced by T helper 1 (Th1) cells which stimulates the activation and proliferation of helper T cells upon released [21]. Various evidences has showed that the protective role of IL-2 against MTB infection were evident, thus higher IL-2 concentration might be observed in subjects with LTBI [21–24]. Meanwhile, IL-10 is a cytokine that can be produced by myeloid cells and was also reported to have a certain correlation with active and latent TB [22, 25, 26]. Until recently, studies to investigate the potencies of these cytokines in LTBI with pregnancy are remained limited. Therefore, we aim to elucidate the biomarker potential of IP-10, IL-2, and IL-10 to determine LTBI status in pregnant women.

## Materials and methods

### Study design and participants

This comparative observational study received ethical approval from The Health Research Ethics Committee, Faculty of Medicine, Universitas Padjadjaran with ethical approval number 994/UN6.C.10/PN/2017. Written informed consent was obtained from all study participants. Subjects with pregnancy status were recruited from Cibabat General Hospital, Ujung Berung General Hospital, Sukabumi General Hospital, and Hasan Sadikin Central General Hospital

**Table 1. Inclusion and exclusion criteria.**

| Inclusion | Exclusion |
|---|---|
| Pregnant women | Non-tuberculosis infection |
| History of exposure with individual with active TB | Gynecological tumor |
| Negative symptom for active TB | Pregnancy complication |
| Negative result for acid-fast staining | |
| Negative culture for *M. tuberculosis* | |

β-HCG: beta-human chorionic gonadotropin; USG: ultrasonography

from December 2017 to July 2018. The inclusion and exclusion criteria for study subjects are described in Table 1. The sample of the study subjects who met the inclusion criteria were then collected and stored at -80˚C refrigerator before processed at The Central Laboratory of Clinical Pathology Department, Hasan Sadikin Central General Hospital.

## QuantiFERON-TB Gold Plus ELISA (QFT-Plus) testing

The IGRA assay, QFT-Plus (Qiagen GmbH, Hilden, Germany) was performed according to manufacturer's instruction [27]. In brief, heparinized sample were equilibrate to room temperature (17–25˚C) prior to transfer to four separate QFT-Plus Blood Collection Tubes: nil tubes containing heparin as negative control, TB Antigen Tube 1 (TB1) containing ESAT-6 and CFP-10 peptide that are designed to elicit CD4+ T-helper lymphocytes, TB Antigen Tube 2 (TB2) containing ESAT-6 and CFP-10 peptide modified to induce CD8+ cytotoxic T lymphocytes, and mitogen tube as a positive control. Next, the aliquoted tubes were inverted to mix for 10 times prior to 16 to 24 hours incubation at 37˚C. After the incubation, the tubes were centrifugated for 15 minutes at 3000 *g* before plasma collection. Next, 50 μL of working strength conjugate, 50 μL of plasma samples, and 50 μL of standards were added to the appropriate ELISA wells according to manufacturer's recommendation. The ELISA plate were then incubated at room temperature for 120 minutes before 6-cycles washing with 400 μL 1X wash buffer and 100 μL addition of substrate solution to each well, followed by incubation at room temperature for 30 minutes. At last, 50 μL of enzyme stopping solution was added to each well. The optical density were measured using an microplate reader at 450 nm filter and a 650 nm reference filter. The results were calculated using QFT Plus analysis software and considered positive if the concentration of TB1 and/or TB2 minus Nil were $\geq$ 0.35 IU/mL or $\geq$ 25% of nil value with nil $\leq$ 8.0 IU/mL and positive mitogen control.

## IP-10, IL-10, and IL-2 determination

The human IP-10, IL-2 and IL-10 ELISA kit were purchased from Elabscience with catalog number E-EL-H0050, E-EL-H6154, and E-EL-H0099, respectively, to determine their concentration level using serum from the study subjects according to manufacturer's instructions [28–30]. Finally, the optical density of IP-10, IL-10, and IL-2 was measured at a wavelength of 450 nm.

## Statistical analysis

The normality test were assessed by using SPSS statistical software (ver.25, SPSS, Chicago, USA). We performed Mann-Whitney U-test to determine the difference of the cytokine levels between IGRA-positive and -negative groups. We considered a *p* value under 0.05 as

**Table 2. Baseline characteristics of the study population.**

| Variable | Total (n = 123) | IGRA | |
|---|---|---|---|
| | | Positive | Negative |
| | | (n = 45) (%) | (n = 78) (%) |
| **Age (years), mean (SD)** | 28±6 | 29±6 | 27±5 |
| **Gestational age** | | | |
| Trimester 1 | 1 (0.8) | 1 (2.2) | 0 (0.0) |
| Trimester 2 | 88 (71.5) | 35 (77.8) | 53 (67.9) |
| Trimester 3 | 34 (27.6) | 9 (20.0) | 25 (32.1) |
| **Parity** | | | |
| Primipara | 62 (50.4) | 16 (35.6) | 45 (57.7) |
| Multipara | 61 (49.6) | 29 (64.4) | 33 (42.3) |
| **Education** | | | |
| Primary school | 6 (4.9) | 1 (2.2) | 5 (6.4) |
| Middle school | 2 (1.6) | 2 (4.4) | - |
| High school | 88 (71.5) | 31 (69.0) | 57 (73.1) |
| College | 20 (16.3) | 9 (20.0) | 11 (14.1) |
| University | 7 (5.7) | 2 (4.4) | 5 (6.4) |

Data are presented as n (%) unless otherwise indicated.

significant. In order to establish a cut-off value for the cytokines, we generated a receiving operating characteristics (ROC) and area under the curve (AUC) by using a webtool from easyROC [31].

## Results

### Baseline characteristics

The demographic and clinical characteristics of study participants are shown in Table 2. Among 123 participants, 45 and 78 subjects were IGRA-positive and -negative with the mean age of 28 years old, respectively. Moreover, 35 out of 45 subjects that belong to IGRA-positive group were under 3rd trimester. In addition, 16 primiparas (35.6%) and 29 multiparas (64.4%) were also tested positive for IGRA.

### IP-10, IL-2, and IL-10 measurement

The comparison of IP-10, IL-2, and IL-10 level between IGRA-positive and IGRA-negative group are shown in Table 3. We demonstrated a highly significant differences in the level of IP-10, IL-10, and IL-2 between those two groups.

Moreover, in the subjects with IGRA-positive, the level of IP-10, IL-10 and IL-2 were found to be increased when compared to IGRA-negative subjects. Interestingly, when the overall

**Table 3. Median concentration of cytokine production in IGRA-positive and -negative groups.**

| Cytokine | IGRA (+) n = 45 | IGRA (-) n = 78 | p-value |
|---|---|---|---|
| | Median (min-max) | Median (min-max) | |
| **IP-10 (pg/mL)** | 1,052.81 (514.48–1282.76) | 605.48 (111.72–646.06) | <**0.001** |
| **IL-10 (pg/mL)** | 8.10 (1.93–129.25) | 5.19 (1.21–121.45) | **0.002** |
| **IL-2 (pg/mL)** | 1,073.42 (70.65–2,440.91) | 902.47 (155.41–2405.05) | **0.008** |

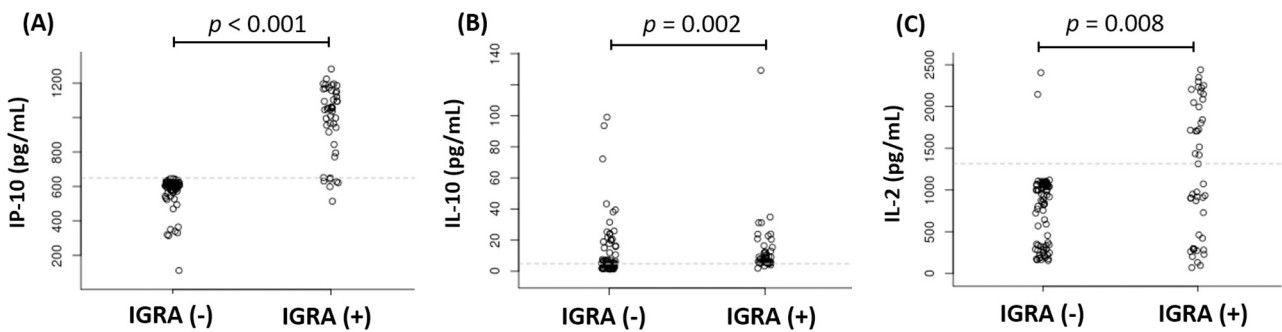

**Fig 1. The overall distribution of (A) IP-10, (B) IL-2, and (C) IL-10 level in IGRA-positive and IGRA-negative groups.**

distribution pattern of IP-10, IL-10, and IL-2 level in IGRA-positive and -negative subjects were visualized by scatter plot as indicated in Fig 1, the best distinction between those subject groups was shown by IP-10.

Furthermore, we analyzed the distribution of cytokine expression by gestational age (Fig 2). Strikingly, IP-10 was found to be able to provide a well-discriminatory value across gestational age with higher concentration level in the 3rd trimester of IGRA-positive subjects (Fig 2A), in contrast to IL-10 and IL-2 where their discriminatory value were only evident in the 2nd trimester (Fig 2B and 2C). Altogether, these results indicated that IP-10 were able to discriminate IGRA-positive group from IGRA-negative group in pregnant women.

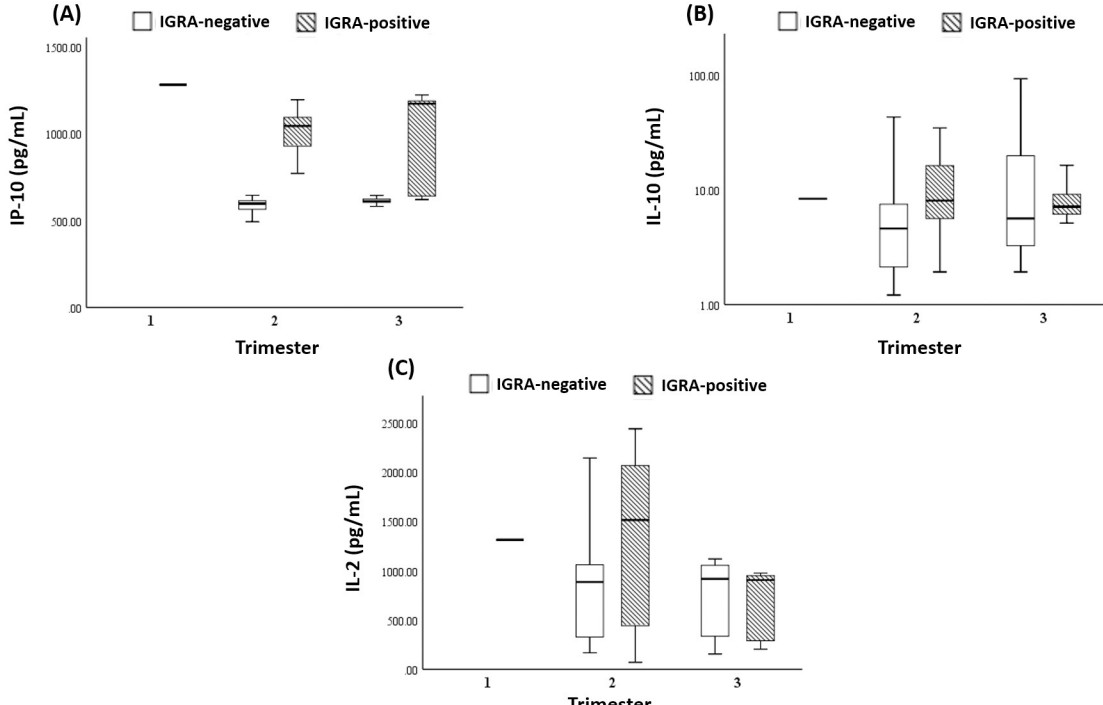

**Fig 2. The distribution of (A) IP-10, (B) IL-2, and (C) IL-10 level in IGRA-positive and IGRA-negative groups by gestational age.**

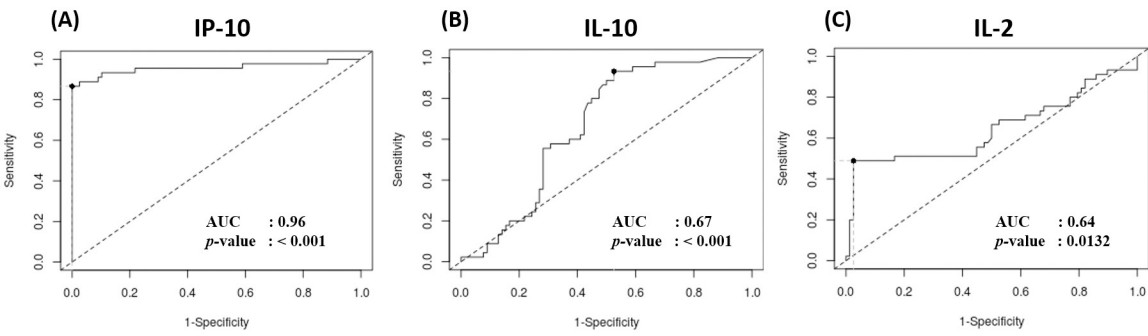

**Fig 3. ROC curve of (A) IP-10, (B) IL-2, and (C) IL-10.**

**Table 4. Performance measurement and cut-off level of IP-10, IL-10, and IL-2.**

| Cytokine | AUC | Cut-off (pg/mL) | Sensitivity | Specificity | PPV | NPV |
|---|---|---|---|---|---|---|
| IP-10 | 0.96 | 649.63 | 0.867 | 1.000 | 1.000 | 0.929 |
| IL-10 | 0.67 | 4.78 | 0.933 | 0.474 | 0.506 | 0.925 |
| IL-2 | 0.65 | 1314.28 | 0.489 | 0.974 | 0.917 | 0.768 |

PPV: positive predictive value; NPV: negative predictive value

### Diagnostic performance of IP-10, IL-10, and IL-2

The diagnostic performance of IP-10, IL-10, and IL-2 was assessed by the ROC curve with the following AUC: 0.96 ($p$-value < 0.001; 95% CI: 0.91–1.00), 0.67 ($p$-value < 0.001; 95% CI: 0.57–0.76), and 0.65 ($p$-value < 0.0132; 95% CI: 0.53–0.76), respectively (Fig 3A–3C). We showed that IP-10, IL-10 and IL-2 produced a relatively high AUC. Moreover, IP-10 produced the greatest AUC and thus had the best capacity in differentiating IGRA-positive from IGRA-negative group.

The optimal cut-off value and diagnostic performance of IP-10, IL-10, and IL-2 are shown in Table 4. Strikingly, our results indicated that with cut-off value of 649.63 pg/mL, IP-10 achieved a sensitivity of 86.7% and specificity of 100%, also a PPV of 100% and NPV of 92.9%; the latter was the highest when compared to IL-10 and IL-2. On the other hand, IL-10 provided the highest sensitivity among these cytokines with 93.3% with a cut-off value of 4.78 pg/mL. Taken together, IP-10 might be the best performing biomarker when compared to IL-10 and IL-2.

### Discussion

In this study, we analyzed the diagnostic performance of IP-10, IL-10, and IL-2 and assessed their potential as alternative biomarkers for IGRA in pregnant women with LTBI. We showed that IP-10 provided the highest discriminatory and diagnostic performance when compared to IL-2 and IL-10 across gestational age.

The pregnancy status could be determined by the dynamic changes in the immune environment [32]. Pro-inflammatory, cell-mediated Th1 responses, marked by the release of interferon gamma (IFN-γ), are required in the early phase of pregnancy, especially for uterine spiral artery remodelling during placenta development [33]. However, elevated level of progresterone and tumor growth factor beta (TGF-β) suppresses Th1 responses and further mediate a progressive shift of immune environment towards anti-inflammatory Th2 responses

[33–35]. As a consequence, the result of IGRA testing, which depends on the production of IFN-γ by Th1 after MTB antigen stimulation might be compromised in immunosuppressive settings [34, 35]. On the other hand, various cytokines that were reported to play a role in active TB could also potentially serve as an alternative biomarker for LTBI in pregnancy [20, 36].

By comparison analysis, we showed that the concentration of IP-10, IL-2, and IL-10 were higher in IGRA-positive pregnant women, suggesting that those cytokines were able to determine LTBI status in pregnant women, while the most significant discriminatory result were showed by IP-10, consistent with its distribution pattern by concentration level when compared to IL-2 and IL-10. Moreover, we further analyzed the distribution of cytokine level by gestational age and found that IP-10 was the only cytokine that showed persistent discriminatory performance across gestational age. Our findings are in concordance with other studies that IP-10 is less dependent on cell-mediated immune response and maintained at higher level even in immunocompromised status compared to IFN-γ [20, 37].

Furthermore, we explored the diagnostic performance of IP-10, IL-10, and IL-2 by using ROC analysis. We revealed that IP-10 showed the highest diagnostic performance with AUC of 0.96 and cut-off of 649.63 pg/mL when compared to IL-10 and IL-2. Strikingly, IP-10 can serve both as a decent predictive and diagnostic biomarker for LTBI in pregnancy with sensitivity and specificity of 86.7% and 100%, respectively. In agreement to previous reports [15, 37, 38], our results provide an additional evidence that IP-10 might serve as a promising biomarker for LTBI in pregnancy.

To our knowledge, our study is the first to investigate the discriminatory and diagnostic performance of IP-10, IL-2, and IL-10 to determine LTBI status in pregnant women in the Indonesian population. Moreover, the utilization of serum of the study subjects and there was no need to conduct MTB antigen stimulation may suggest that the determination of cytokine level was easier to implement than IGRA.

However, our study has certain limitations. First, the gestational age data of the study subjects were not well-spread where the first semester was consist of only one subject. That could be due to randomized nature of subject selection where almost all of the study subjects were already in the second or third semester when the pregnancy status was revealed. Second, there is no gold standard for LTBI diagnosis, providing a consistent challenge for the development of novel biomarkers for LTBI.

Nevertheless, the diagnostic biomarkers for LTBI in pregnancy are still limited. However, we are able to demonstrate that IP-10 is a highly reliable cytokine biomarker to be used to diagnose LTBI in pregnant women across gestational ages.

## Conclusion

As a conclusion, our findings suggest that IP-10 has a highly diagnostic potential for LTBI in pregnant women when compared to IL-10 and IL-2. Therefore, we encourage that IP-10 can serve as an alternative cost-effective biomarker to screen and diagnose latent tuberculosis in pregnancy across gestational ages.

## Supporting information

**S1 Table. Raw data for IP-10, IL-2, and IL-10 level.**
(XLSX)

## Acknowledgments

We sincerely thank the laboratory staff of Department of Clinical Pathology, Faculty of Medicine, Universitas Padjadjaran/Hasan Sadikin Central Hospital for their tremendeous technical assistance.

## Author Contributions

**Conceptualization:** Agnes Rengga Indrati, Anton Sumarpo, Rositha Ratna Wisesa, Mohammad Ghozali, Raden Tina Dewi Judistiani, Budi Setiabudiawan.

**Data curation:** Agnes Rengga Indrati, Petty Atmadja, Rositha Ratna Wisesa, Budi Setiabudiawan.

**Formal analysis:** Agnes Rengga Indrati, Petty Atmadja, Rositha Ratna Wisesa, Raden Tina Dewi Judistiani.

**Funding acquisition:** Anton Sumarpo, Mohammad Ghozali, Raden Tina Dewi Judistiani, Budi Setiabudiawan.

**Investigation:** Anton Sumarpo, Petty Atmadja, Rositha Ratna Wisesa, Mohammad Ghozali, Raden Tina Dewi Judistiani, Budi Setiabudiawan.

**Methodology:** Agnes Rengga Indrati, Petty Atmadja, Rositha Ratna Wisesa, Mohammad Ghozali, Raden Tina Dewi Judistiani, Budi Setiabudiawan.

**Project administration:** Anton Sumarpo, Mohammad Ghozali, Raden Tina Dewi Judistiani, Budi Setiabudiawan.

**Resources:** Petty Atmadja, Rositha Ratna Wisesa.

**Software:** Agnes Rengga Indrati, Petty Atmadja, Rositha Ratna Wisesa.

**Supervision:** Anton Sumarpo, Mohammad Ghozali, Raden Tina Dewi Judistiani, Budi Setiabudiawan.

**Validation:** Agnes Rengga Indrati, Anton Sumarpo, Mohammad Ghozali, Raden Tina Dewi Judistiani, Budi Setiabudiawan.

**Visualization:** Agnes Rengga Indrati, Petty Atmadja.

**Writing – original draft:** Agnes Rengga Indrati, Anton Sumarpo, Petty Atmadja, Rositha Ratna Wisesa.

**Writing – review & editing:** Agnes Rengga Indrati, Anton Sumarpo, Petty Atmadja, Rositha Ratna Wisesa, Mohammad Ghozali, Raden Tina Dewi Judistiani, Budi Setiabudiawan.

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
