## [Decision Letter · Decision Letter 0]

28 May 2022

PONE-D-22-12130Exploring alternative cytokines as potential biomarkers for latent tuberculosis infection in pregnant womenPLOS ONE

Dear Dr. Anton Sumarpo,

Thank you for submitting your manuscript to PLOS ONE. After careful consideration, we feel that it has merit but does not fully meet PLOS ONE’s publication criteria as it currently stands. Therefore, we invite you to submit a revised version of the manuscript that addresses the points raised during the review process.

We look forward to receiving your revised manuscript.

Kind regards,

Wenping Gong, Ph.D.

Academic Editor

PLOS ONE

Journal Requirements:

"No"

3. Please amend the manuscript submission data (via Edit Submission) to include author Agnes Rengga Indrati

5. Please ensure that you refer to Figure 3 in your text as, if accepted, production will need this reference to link the reader to the figure.

Reviewers' comments:

Reviewer's Responses to Questions

**Comments to the Author**

1. Is the manuscript technically sound, and do the data support the conclusions?

Reviewer #1: Yes

Reviewer #2: Yes

2. Has the statistical analysis been performed appropriately and rigorously? 

Reviewer #1: Yes

Reviewer #2: I Don't Know

3. Have the authors made all data underlying the findings in their manuscript fully available?

Reviewer #1: Yes

Reviewer #2: Yes

4. Is the manuscript presented in an intelligible fashion and written in standard English?

Reviewer #1: Yes

Reviewer #2: Yes

5. Review Comments to the Author

Reviewer #1: The topic selection of this article has certain theoretical significance and application value. The current situation and problems of tuberculosis latent infection in pregnancy are summarized, and explored the possibility of IP-10, IL-10 and IL-2 to judge LTBI in pregnancy, with certain innovation and operability, and the research direction is clear.The article is written standard, the content is relatively complete, the statistical method is correct, the chart drawing is qualified,prominent focus, has certain clinical significance.The drawback is that the comparison between pregnant women and the general population is not made, and the sample size is also small, and the persuasion is limited.

Reviewer #2: This manuscript represented an interesting piece of work about the diagnosis of LTBI in pregnant women. Some minor flaws have been found and listed below after reviewed the manuscript. I suggest to accept the manuscript in the reivised form.

1. Why there are two age categories listed in table 2? The result of the diagnostic biomarkers for LTBI in pregnancy has nothing to do with the age categories. Furthermore, the data of the two age categories is not used for analysis in this paper. It is recommended not to group age categories here.

2. As the author said: “The gestational age data of the study subjects were not well-spread where the first semester was consist of only one subject.” Therefore, more pregnant women in trimester 1 should be involved in this study.

6. PLOS authors have the option to publish the peer review history of their article (what does this mean?). If published, this will include your full peer review and any attached files.

Reviewer #1: No

Reviewer #2: No

---

## [Author Response · Author response to Decision Letter 0]

10 Jun 2022

A. Comments from academic editor:

Response: We are thankful for your input. Therefore, we have revised the file name according to the guidelines. 

2. Thank you for stating the following in your Competing Interests section: "No". Please complete your Competing Interests on the online submission form to state any Competing Interests. If you have no competing interests, please state "The authors have declared that no competing interests exist.", as detailed online in our guide for authors at http://journals.plos.org/plosone/s/submit-now

Response: We are thankful for your support. In addition, we also revise the competing interest declaration sentence in the cover letter to "The authors have declared that no competing interests exist."

3. Please amend the manuscript submission data (via Edit Submission) to include author Agnes Rengga Indrati

Response: We are thankful for your comment. Therefore, we have checked and made sure that Agnes Rengga Indrati is included as one of the authors and also as corresponding author as indicated in a screenshot picture shown in rebuttal letter. 

Response: We are thankful for your valuable comment. We have checked and made sure that our ethics statement is only appear in the Methods section of our manuscript. In addition, we also removed the attached letter that consists of ethical statement from the uploading section in editorial manager platform.

5. Please ensure that you refer to Figure 3 in your text as, if accepted, production will need this reference to link the reader to the figure.

Response: We are thankful for your valuable comment. We have revised the manuscript to refer to Figure 3 in the manuscript text. 

Response: We are thankful for your valuable comment. We have checked the reference list and ensured that all references are correct and not retracted. 

B. Comments from reviewers:

1. Reviewer #1: The topic selection of this article has certain theoretical significance and application value. The current situation and problems of tuberculosis latent infection in pregnancy are summarized, and explored the possibility of IP-10, IL-10 and IL-2 to judge LTBI in pregnancy, with certain innovation and operability, and the research direction is clear. The article is written standard, the content is relatively complete, the statistical method is correct, the chart drawing is qualified, prominent focus, has certain clinical significance. The drawback is that the comparison between pregnant women and the general population is not made, and the sample size is also small, and the persuasion is limited.

Response: We are grateful for your valuable comments. We acknowledge that our study had certain limitations, including small sample size. On the other hand, due to the difference in immunological condition between pregnant and general population, especially cytokine expression that may result in diverse variability, we only included pregnant women population in our study.

2. Reviewer #2: This manuscript represented an interesting piece of work about the diagnosis of LTBI in pregnant women. Some minor flaws have been found and listed below after reviewed the manuscript. I suggest to accept the manuscript in the reivised form.

1. Why there are two age categories listed in table 2? The result of the diagnostic biomarkers for LTBI in pregnancy has nothing to do with the age categories. Furthermore, the data of the two age categories is not used for analysis in this paper. It is recommended not to group age categories here. 

2. As the author said: “The gestational age data of the study subjects were not well-spread where the first semester was consist of only one subject.” Therefore, more pregnant women in trimester 1 should be involved in this study.

 Response: 

1. We are grateful for your valuable input; therefore, we will revise the manuscript by deleting group age categories. 

2. We acknowledge that the gestational age data is the limitation of this study; however, the subject for this experiment were collected randomly to avoid population bias. Consequently, there was a chance that we may acquire lesser samples from trimester 1 subject.

---

## [Editor Report · Decision Letter 1]

14 Jun 2022

Exploring alternative cytokines as potential biomarkers for latent tuberculosis infection in pregnant women

PONE-D-22-12130R1

Dear Dr. Agnes Rengga Indrati,

We’re pleased to inform you that your manuscript has been judged scientifically suitable for publication and will be formally accepted for publication once it meets all outstanding technical requirements.

Kind regards,

Wenping Gong, Ph.D.

Academic Editor

PLOS ONE

---

## [Editor Report · Acceptance letter]

30 Jun 2022

PONE-D-22-12130R1 

Exploring alternative cytokines as potential biomarkers for latent tuberculosis infection in pregnant women 

Dear Dr. Indrati:

I'm pleased to inform you that your manuscript has been deemed suitable for publication in PLOS ONE. Congratulations! Your manuscript is now with our production department. 

Kind regards, 

on behalf of

Dr. Wenping Gong 

Academic Editor

PLOS ONE